# Post-Transcriptional and Post-Translational Modifications in Telomerase Biogenesis and Recruitment to Telomeres

**DOI:** 10.3390/ijms24055027

**Published:** 2023-03-06

**Authors:** Nikita Shepelev, Olga Dontsova, Maria Rubtsova

**Affiliations:** 1Shemyakin-Ovchinnikov Institute of Bioorganic Chemistry, Russian Academy of Sciences, Moscow 117437, Russia; 2Chemistry Department and Belozersky Institute of Physico-Chemical Biology, Lomonosov Moscow State University, Moscow 119234, Russia; 3Skolkovo Institute of Science and Technology, Moscow 121205, Russia

**Keywords:** telomerase, modifications, telosome, biogenesis, processing, telomerase RNA, TERT, hTERP

## Abstract

Telomere length is associated with the proliferative potential of cells. Telomerase is an enzyme that elongates telomeres throughout the entire lifespan of an organism in stem cells, germ cells, and cells of constantly renewed tissues. It is activated during cellular division, including regeneration and immune responses. The biogenesis of telomerase components and their assembly and functional localization to the telomere is a complex system regulated at multiple levels, where each step must be tuned to the cellular requirements. Any defect in the function or localization of the components of the telomerase biogenesis and functional system will affect the maintenance of telomere length, which is critical to the processes of regeneration, immune response, embryonic development, and cancer progression. An understanding of the regulatory mechanisms of telomerase biogenesis and activity is necessary for the development of approaches toward manipulating telomerase to influence these processes. The present review focuses on the molecular mechanisms involved in the major steps of telomerase regulation and the role of post-transcriptional and post-translational modifications in telomerase biogenesis and function in yeast and vertebrates.

## 1. Introduction

Telomerase provides eukaryotic cells unlimited proliferative potential by maintaining their telomeres [1,2]. Telomeres shorten with each division of somatic cells because of the end-replication problem [3,4] and endonuclease action [5]. When telomeres become critically shortened, they cannot protect the ends of linear chromosomes from DNA damage signaling, and the affected cells become subject to DNA damage arrest and/or senescence [6,7]. However, some cells manage to survive by activating telomerase to elongate telomeres. This phenomenon has been observed in the majority of cancer cells [8]. Telomerase is active in cells that should divide continuously during the life of an organism, and it is activated in special cases in which extended proliferation is required [9,10]. Immune cells activate telomerase during differentiation and activation [11,12,13]. Regenerative processes also require telomerase activation [14]. Telomerase action is important, as it preserves telomere length during embryonal development [15,16]. A deficiency in telomerase activity during development and in stem cells is associated with diseases related to premature senescence because of the decreased regenerative capacity of the affected stem cells [17]. Only a minority of cancer cells can undergo the alternative lengthening of telomeres (ALT) without the involvement of telomerase [18].

Telomerase RNA and telomerase reverse transcriptase are two major components of the telomerase complex (or telomerase holoenzyme), which are sufficient for the telomerase activity in vitro [19,20,21]. However, the telomerase holoenzyme’s assembly and interaction with telomeres require many additional components. Hereinafter, we use the terms “telomerase complex” or “telomerase holoenzyme” to refer to a fully assembled and catalytically active telomerase ribonucleoprotein (RNP) complex. “Telomerase RNP” is used as an immature telomerase ribonucleoprotein complex, which lacks some auxiliary proteins unless otherwise specified. The deficiency and mutations of additional telomerase components affect the telomerase action of telomeres. The biogenesis of these components is regulated at every step, from transcription and processing to maturation and the posttranslational modifications of telomerase reverse transcriptase and telomerase auxiliary proteins. It is essential to understand the regulatory mechanisms of telomerase biogenesis and activity to develop approaches with which to manipulate telomerase, thereby influencing regeneration, maintaining fitness, and preventing senescence and cancer progression.

The present review focuses on the molecular mechanisms that regulate the major steps of biogenesis and the function of telomerase components in yeast and vertebrates.

## 2. Yeast Telomerase

### 2.1. Yeast Telomerase Holoenzyme Composition

The main components of budding yeast telomerase complex are telomerase RNA (TLC1) and telomerase reverse transcriptase (Est2). Est1, Est3, Pop1, Pop6, Pop7, the yKu70/80 heterodimer, and the heptameric Sm_7_ protein ring are also part of the telomerase holoenzyme and are relatively stably bound to TLC1 via a set of protein–RNA and protein–protein interactions (Figure 1).

Although in vitro telomerase activity requires only TLC1 and Est2 [19], the listed auxiliary proteins are absolutely indispensable for the full activation of telomerase in living cells. Sm_7_ is required for the stabilization of the cytoplasmic pool of TLC1 by binding near the 3′ end of TLC1 in a structure called a terminal arm or Sm arm [22,23]. Est2 binds to the pseudoknot and the template region in a proximal center of TLC1 where three large TLC1 stems meet [24]. Est1 mediates the major pathway of telomerase recruitment to telomeres in the late S phase, as it can bind Cdc13 telomeric protein and telomerase RNA [25,26,27,28,29,30]. Moreover, Est1 participates in the activation of telomerase at telomeres, changing the Cdc13 conformation and influencing its ability to retain the telomeric 3′ end and hide it from telomerase [31]. Est1 binding relies on hinge–hairpin and bulge elements in a part of TLC1 known as an Est1 arm [32]. The yKu70/80 heterodimer is a key factor of DNA break repair because of its ability to specifically associate with double-stranded DNA breaks. However, its DNA binding domain allows it to recognize a short stem-loop structure within another TLC1 arm (yKu arm) [33,34,35]. In addition, the yKu80 protein can bind Sir4 (silent information regulator 4) telomeric protein, thus providing telomerase with an alternative means of finding the telomeric end [35,36]. Perhaps the least understood protein within the telomerase holoenzyme is Est3, which is a small, oligonucleotide/oligosaccharide-binding fold (OB-fold) containing protein whose binding is thought to be required for the correct conformation of the complex, potentially regulating the telomerase assembly pathway [37]. The Est3 protein binds to the complex via interactions with Est1 and Est2 [37]. OB-fold proteins are a hallmark of an important evolutionarily conserved subclass of proteins that are involved in maintaining the integrity of the genome, particularly with respect to telomeres [38]. Finally, the telomerase complex contains a set of Pop proteins (Pop1, Pop6, and Pop7), which it shares with the RNase P and RNase MRP complexes [39]. Pop proteins bind to the CS2a/TeSS domain of the Est1 arm near the Est1 binding site. The presence of Pop proteins was found to be important for the biogenesis of telomerase and its nuclear localization [40,41]. It is worth noting that telomerase is an extremely lowly abundant complex (~30 of TLC1 molecules per cell), which complicates the comprehensive investigation of its composition [42]. More than 100 proteins were found to co-precipitate with overexpressed Est1 and Est2 subunits in a study conducted by Lin et al., suggesting that other telomerase holoenzyme components might be revealed in future [43].

### 2.2. Telomerase RNA Processing in Yeast

In the late G1 phase of the cell cycle, RNA polymerase II synthesizes a ~1220 nt primary TLC1 transcript, which is subsequently processed, giving rise to the mature 1157 nt poly(A) form [44,45]. Nrd1/Nab3-dependent transcription termination was found to be crucial for the generation of mature TLC1, thus mirroring the maturation pathway of the snRNA transcripts in budding yeast [45,46]. The Nrd1 protein can bind to the phosphorylated C-terminal domain (CTD) of RNA polymerase II, with a preference for the phospho-Ser5 form of the CTD repeat. At a later stage of transcription, Nrd1 forms a heterodimer with the Nab3 protein, and they recognize motifs located near the mature 3′ end of TLC1. The termination of TLC1 transcription most likely ends with the concomitant exonucleolytic cleavage of TLC1 [45,46]. However, ~10% of cellular TLC1 is present in a longer ~1240 nt form (plus a ~80 nt poly(A) tail), which is likely generated via the termination near the A/U-rich sequences akin to the majority of mRNA species [45,47]. The potential functional roles of the longer poly(A)+ species (e.g., whether it serves as a precursor to the 1157 nt poly(A) form) have yet to be established; it could be simply generated as a by-product of a redundant “fail-safe” mechanism of transcription termination [45,48]. At the 5′ end, TLC1 is capped by 7-methylguanosine, which is later converted into the 5′-2,2,7-trimethylguanosine (TMG) cap, thereby resembling snRNA maturation [22].

Interestingly, the 3′-processing mechanism of *S. cerevisiae* may be a notable exception within the fungi kingdom. First discovered in a fission yeast species *Schizosaccharomyces pombe* [49], the spliceosomal cleavage reaction was shown to be an alternative pathway for telomerase RNA maturation in filamentous fungi (*Neurospora crassa* [50] and different *Aspergillus* species [51]). Even within the budding yeast clade, telomerase RNAs from distantly related branches were found to contain conserved intronic elements downstream of the mature 3′ end positions [52,53,54], and the mutational analysis of telomerase RNA (TER) from *Hansenula polymorpha* confirms the importance of these sequences for the RNA accumulation [53].

### 2.3. Assembly of the Yeast Telomerase RNA–Protein Complex

Although there is no experimental evidence to support this supposition, the newly synthesized TLC1 is most likely bound by the cap-binding complex (CBC) and transcription–export complex 1 (TREX-1), as is the case for other m^7^G-capped RNAs [55,56]. These complexes are thought to provide assistance during the early steps of TLC1 maturation, including the recruitment of the Xpo1 exportin. Data suggest that TLC1 cytoplasmic export is entirely dependent on Xpo1 [23,57]. However, the export receptor heterodimer Mex67-Mtr2 likely serves as an adaptor during export and a stabilizer of RNA, as in *mex67-5* mutants cells, TLC1 is rapidly degraded by the nuclear exosome [23,58].

After its exportation to the cytoplasm, the Sm_7_ heptameric ring is assembled around the sequence near the 3′ end of TLC1 [55]. Contrary to the earlier proposal [57], the Sm complex is not required to ensure RNA stability immediately after transcription, thus allowing TLC1 molecules with mutations within the Sm site to reach the cytoplasm in an intact form [23]. Slight variations in 3′ end protection mechanisms exist among fungi; for instance, only the precursor form of *S. pombe* telomerase RNA (TER1) is bound by the Sm_7_ complex, and its substitution by a Lsm2-8 heptameric ring is required for the stability of the mature RNA [59]. In addition, the evolutionarily conserved Lar7 protein (also known as Pof8) is important for the stabilization of the binding of the Lsm protein to TER1 [60]. Thc1 and Bmc1 work cooperatively with Lar7 to recognize correctly folded TER1 and promote the recruitment of the Lsm2-8 heptamer [61,62]. We should note that fission yeasts highly diverged from budding yeasts [63], particularly in the context of telomere protection, as discussed below.

In the absence of any Est proteins, *S. cerevisiae* telomerase RNA accumulates in the cytoplasm, indicating that the assembly of the active RNP takes place in the cytoplasm and is required for re-import into the nucleus [57]. However, these data are not inconsistent with the binding of (at least one of) the Est proteins in the nucleus; thus, the exact timing of the addition of each Est protein is unclear [55]. A study conducted by Tucey and Lundblad [37] revealed a complex regulated process, with the Est3 subunit acting as a switch controlling the assembly/disassembly pathways of RNP. Est2 and Est1 can directly bind specific regions of telomerase RNA [24,25,26,64], and the accumulation of the Est1–TLC1–Est2 subcomplex can be detected early in the cell cycle [37]. The existence of the complete telomerase complex containing all three Est subunits appears to be transient, as closer to the end of the cell cycle, the Est1–TLC1–Est3 subcomplex accumulates [37]. The dissociation of Est2 in the G2/M phases may be explained by its interaction with another protein. PinX1 protein, which is reported to be an Est2 arrest factor in the nucleolus, fits this role [65].

Experiments with alternative budding yeast systems underscore the additional complexities within the suggested models of telomerase assembly. Perhaps the most prominent example is the absence of *EST1* genes from the genomes of *Candida parapsilosis* and *Lodderomyces elongisporus* and the increased size of their *EST3* open reading frames (ORFs), which were shown to play an important role in the interaction between Est3 and Est2 [66]. On the other hand, there are *Candida albicans* and *H. polymorpha,* which are apparently completely dependent on the Est1–TLC1 interaction during the prior binding of Est3 (thus suggesting an inability to form the Est1–TER–Est2 subcomplex) [67,68]. In addition, the loss of either Est1 or Est3 leads to the strong destabilization of the telomerase RNP and the degradation of the catalytic subunit in *H. polymorpha* [68]. Deciphering which of the described inconsistencies reflect different sides of the same process or the evolutionary plasticity of the telomerase assembly mechanisms could be an interesting challenge for future studies.

The additional stabilization of the ‘Est-bound’ telomerase RNA was found to require the binding of Pop proteins to the CS2a/TeSS domain of TLC1. In their absence (in *pop1* and *pop6* mutants), Est1–TLC1 binding is reduced, and telomerase RNA accumulates in the cytoplasm [40,41,69]. As one of the essential RNA-processing machineries, Pop proteins are present in most organisms, and elements similar to the CS2a/TeSS of TLC1 can be discerned in telomerase RNAs from many yeasts (including *S. pombe* and *H. polymorpha* [39]). It has yet to be determined whether Pop proteins also interact with Est3 and influence its association with the telomerase complex. This may represent an important topic to study. In contrast, the binding of another telomerase subunit—the yKu70/80 heterodimer—may be specific to *S. cerevisiae* and its close relatives [53,54,70]. Consistent with this idea, a stable association between yKu70/80 and TER (TLC1 orthologue) in an RNA co-immunoprecipitation experiment was not observed in *H. polymorpha* [71]. This may be reflected by the fact that even in *S. cerevisiae*, the role of the yKu70/80–TLC1 interaction is relatively minor, and it seems to play a secondary role in telomerase recruitment to telomeres compared to the major Cdc13–Est1 pathway [35]. However, the binding of Ku to TLC1 may be important for the robust import of the telomerase RNP into the nucleus, as TLC1 molecules accumulate in the cytoplasm in the *∆yku70* mutant [57].

Irrespective of the exact composition and conformation of the telomerase RNP, the importin Mtr10 and karyopherins Kap122 and Cse1 were implicated in the nuclear import of telomerase [57,69,72]. However, their importance is debatable, since the *mtr10* knock-out strain has a pleiotropic phenotype and low TLC1 levels [72], and the effect of *∆kap122* mutation was found to be minor in another study [23]. The final step of telomerase maturation is the addition of a TMG cap at the 5′ end of TLC1 and the 3′ end’s trimming by the exosome, which most likely happen in the nucleolus after the import into the nucleus of the fully assembled telomerase complex [23,69]. Finally, it is worth mentioning that the existing data are not inconsistent with the repeated shuttling of telomerase between the nucleus and the cytoplasm; this possibility was discussed by Bartle et al. in 2021 [55]. In Figure 2, we compiled a general scheme of the telomerase complex’s biogenesis in budding yeast.

### 2.4. Telomerase Recruitment to Telomeres in Yeast

Budding yeast telomeres are protected by a telomere chromatin structure called telosome (Figure 2), which is distinct from the structure found in metazoans and fission yeast. Fission yeast protects telomeres using orthologues of vertebrate telomeric proteins [73,74], which are discussed below. The major telosome protein is repressor/activator protein 1 (Rap1) [75]. Rap1 binds to telomeric double-stranded DNA, as well as Rif1 and Rif2 proteins (Rap1-interacting factors 1 and 2), which negatively regulate telomere length [76,77]. Rif1 and Rif2 also compete with the Sir2, Sir3, and Sir4 proteins for association with Rap1. Sir2/Sir3/Sir4 mediate gene silencing at the telomere [78,79]. Additionally, a telomere-specific RPA-like complex, which contains Cdc13/Stn1/Ten1 (the CST complex), binds to telomeric single-stranded DNA [80]. Hrq1 and Pif1 helicases can also bind to telomeres and play a dual role in telomere length homeostasis [81,82]. On the one hand, they can remove telomere structures such as G-quadruplexes, thereby promoting telomerase recruitment [83]. On the other hand, they can unwind the DNA–RNA hybrid formed by the 3′ end of telomeres and TLC1, thereby inhibiting telomere elongation [84,85].

Telomerase recruitment to telomeres in budding yeast occurs via two different mechanisms: the Sir4–yKu80 interaction, which mostly occurs in the G1 phase of the cell cycle, and the Cdc13–Est1 interaction, which is the prevailing recruitment pathway during the late S phase, when telomere elongation takes place [86]. Disrupting the Sir4–yKu80 pathway results in only mild telomere shortening, highlighting the critical role of the Cdc13–Est1 interaction during S phase as the primary functional pathway for telomerase recruitment [35,87].

Telomerase function is also limited by telomere length, as telomerase preferentially targets short telomeres [88]. The preference for short telomeres is mediated by the Rap1-interacting partners Rif1 and Rif2. Together, they form a negative feedback loop that regulates telomere elongation in a length-dependent manner [89,90]. The Rap1/Rif1/Rif2 regulatory mechanism relies on the number of Rif proteins associated with a telomere as an indicator of individual telomere length. Thus, only telomeres with low concentrations of Rif1 and Rif2 will be elongated.

Late S phase telomeres have a notable characteristic that distinguishes them from G1 and G2 chromosome ends, namely, the formation of detectable 3′ single-stranded telomeric overhangs [91]. The formation of telomeric overhangs requires 5′ end processing in late S phase after the passage of the replication fork [92]. This process is largely reliant on the Mre11–Rad50–Xrs2 (MRX) complex with the assistance of the Sae2 protein [93,94]. However, the MRX complex may also play a structural role in telosome maintenance [95].

The checkpoint kinase Tel1 preferentially localizes to short telomeres and also mediates telomerase’s preference for short telomeres [96,97]. The balance between Tel1 and Rif2 activities determines the extent of telomere processing via the MRX complex [98]. On the one hand, Tel1 enhances MRX-dependent 5′ telomere processing, while Rif2, on the other hand, inhibits MRX activity [98,99,100]. The mechanism of counting Rif proteins, which ultimately targets telomerase to short telomeres, relies on an elaborate network of physical and functional interactions between the Rif1 and Rif2 proteins, Tel1, and the MRX complex.

There is some evidence that Cdc13 is phosphorylated by Tel1 and Mec1 checkpoint kinases (orthologues of ATM in metazoans) to promote the Cdc13–Est1 interaction in the S phase [101,102]. In turn, the action of PP2A phosphatase and Aurora kinase on Cdc13 limits telomere elongation during the G2/M phases [102]. Interestingly, fission yeast orthologues of ATM and ATR also promote the recruitment of telomerase to telomeres [103]. The Cdc13–Est1 interaction permits telomerase recruitment, while the CST complex prohibits it to prevent the excessive elongation of telomeres [104]. The switch between these two complexes might serve as an additional regulatory mechanism of telomerase’s function with respect to telomeres. Thus, post-translational modifications of telomeric proteins also play an important role in the recruitment of yeast telomerase to telomeres. Some other modifications are briefly discussed in [105].

### 2.5. Post-Translational Modifications (PTMs) of the Telomerase Components in Yeast

Despite several decades of investigation into yeast telomerase, only one PTM of a telomerase subunit has been studied (in the context of telomerase function): the ubiquitination of the Est1 protein from *S. cerevisiae*. The amount of Est1 is regulated during the cell cycle (with a peak in S phase), and proteasomal degradation is considered to be responsible for this [106,107,108]. In the related studies, the addition of ubiquitin by the Ufd4 E3 Ub ligase was detected, while the while the cdc48-3 mutation (a component of a complex targeting proteins to the proteasome) and the ufd4 deletion increased the amount of Est1; this affected telomerase assembly and telomere homeostasis [43].

The possibility of Est3’s phosphorylation was suggested by Tuzon et al. [109]. However, which residue could be modified, and by which kinase, and the potential functional consequences of such a modification were not identified.

Several papers have reported the PTM of the yKu70/80 heterodimer with small ubiquitin-like modifiers (SUMO) or through yKu70/80 SUMOylation. Three SUMO E3 ligases (Mms21, Siz1, and Siz2) were implicated in the modification of the C-terminus of yKu70 [110,111,112], while it was also determined that Siz2 can also modify yKu80 [111]. Although the disruption of yKu70/80 SUMOylation leads to significant changes in telomere length, these changes were not linked to problems with the telomerase assembly process. Defects in telomere silencing and anchoring were also described; however, the effects caused by mutations in SUMO E3 enzymes are difficult to interpret, as multiple proteins are affected by SUMOylation (including several telomeric proteins [113]). Finally, the phosphorylation of the Ser623 residue of yKu80 by the Pho85 kinase was discovered, but telomere maintenance does not seem to be perturbed by the yku80S623A mutation [114].

Notably, a number of post-translational modifications in telomerase proteins were identified during several genome-wide screenings [115,116,117]. Although the existence of these PTMs must be carefully confirmed, it would be interesting to study their potential involvement in telomere maintenance. Considering the crucial roles PTMs play in the control of diverse cellular processes and the fact that, so far, only one has been implicated in telomerase biogenesis, it is likely that some of the mentioned (or yet unidentified) modifications will be found to regulate the assembly of the yeast telomerase holoenzyme.

## 3. Telomerase in Vertebrates

### 3.1. Human Telomerase Holoenzyme Composition

The biogenesis of the vertebrate telomerase complex has been most extensively studied in humans, which is largely due to the medical significance of the appropriate functioning of telomerase. Therefore, the results obtained primarily from human cells will be considered further. One well-known, rare hereditary disease associated with the downregulation of telomerase activity is dyskeratosis congenita (DC) [118]. The disruption of the functioning of the protein dyskerin (DKC1) leads to a decrease in the content of telomerase in cells and the shortening of telomeres. Point mutations in dyskerin lead to the formation of an X-linked form of DC [119], in which a drop in telomerase level is accompanied by defects in actively proliferating tissues, bone marrow, lungs, and skin. Autosomal dominant forms of DC [120] have been observed with mutations in telomerase reverse transcriptase, telomerase RNA [121], and telomeric protein TIN2 [122]. These diseases indicate a direct and important link between the pathophysiology of DC and telomere shortening.

Human telomerase RNA (hTR) and human TERT (hTERT) are sufficient for the generation of telomerase activity in vitro in rabbit reticulocyte lysate, which provides accessory proteins for the assembly of the telomerase complex [20,21]. However, the effective functioning of the telomerase complex in vivo requires many additional proteins [123]. Thus, we briefly consider the secondary structure of telomerase RNA and the domain organization of telomerase reverse transcriptase (Figure 3A,B). For more details on the structure of the human telomerase holoenzyme (Figure 3C), the reader may refer to a recent review [124].

Phylogenetic comparison of telomerase RNAs among vertebrates has identified several conserved regions, including a pseudoknot, CR4/5 domain, and H/ACA domain [125]. Other important elements include the template and template boundary element (TBE) (Figure 3A).

In most organisms, including vertebrates, TERT contains four domains: the telomerase essential N-terminal domain (TEN), telomerase RNA-binding domain (TRBD), reverse transcriptase domain (RT), and the C-terminal extension domain (CTE) [126] (Figure 3B). The amino acid linker between the TRBD and TEN domains is a low-complexity proline/arginine/glycine-rich region that may promote TERT dimerization or may be a site of protease cleavage in human cells [127] (Figure 3B, shown in gray).

### 3.2. hTR Biosynthesis and Early Processing

Human telomerase RNA is synthesized by RNA polymerase II in the form of a precursor elongated at the 3′ end [128,129,130] and monomethylated at the 5′ cap [131]. After numerous stages of processing, human telomerase RNA turns into a mature form consisting of 451 nucleotides, which makes up about 70% of the total hTR in a human cell [132]. Mature hTR differs in many ways from processed mRNA, which is also transcribed by RNA polymerase II. For example, mature telomerase RNA does not have a poly(A) tail [133] and also contains a trimethylated 5′ cap [131]. Several studies have demonstrated the presence of different forms of immature hTR; however, establishing their biological role requires further study [132,134]. It was demonstrated that a primary transcript of hTR elongated at the 3′ end primary region is transported into the cytoplasm [135] and translated into a protein named hTERP (human Telomerase RNA Protein) [136]. hTERP protects cells from apoptosis and regulates autophagy through the modulation of the activity of AMPK and TSC2 kinases [137]. Moreover, another study recently revealed the import of hTR in mitochondria where it is processed into a TERC-53 product, which is then re-exported into the cytoplasm [138]. The level of TERC-53 in the cytoplasm responds to mitochondrial function and plays a regulatory role in cellular senescence [139].

Near the 5′ end of the hTR transcript is a series of guanosines forming a G-quadruplex that protects hTR from degradation at the beginning of transcription [140]. The DHX36 RNA helicase (also known as RHAU) binds and resolves the G-quadruplex, which contributes to the correct folding of telomerase RNA and the formation of the P1 helix in the template boundary element (TBE) [141,142] (Figure 3A). In subsequent work, it was demonstrated that the heterogeneous nuclear ribonucleoproteins F, H1, and H2 (hnRNP F/H complex), which regulate alternative splicing by binding G-rich RNA sequences [143,144], also interact with the G-rich region at the 5′ end of hTR [145]. hnRNP F/H is assumed to contribute to TBE stabilization due to the preferential binding of the G-rich region without the folding of the G-quadruplex [145].

Mediator and Integrator are multi-subunit complexes that serve as links between specific transcription factors and RNA polymerase II bound to common transcription factors. Mediator and Integrator coordinate effective transcription by RNA polymerase II [146,147]. Mediator is necessary for the formation of a pre-initiation complex for the transcription of most mRNAs [146]. In turn, the Integrator complex is responsible for regulating the transcription of non-coding RNAs [147]. Recent work has shown that the termination of the transcription of the hTR occurs with the assistance of the Integrator complex [148]. The depletion of Integrator subunits results in the accumulation of elongated hTR transcripts.

Elongated hTR transcripts are processed into a mature form, degraded in a competing manner, or carry out alternative functions [133,136,138,148]. The hTR-processing steps were discussed in further detail in a recent review [149].

### 3.3. H/ACA Motif Pre-Assembly of Telomerase RNP

Biogenesis and hTR accumulation do not require the participation of hTERT [150]. The correct processing of telomerase RNA requires the presence of an H/ACA motif. The H/ACA motif has a conservative secondary structure that consists of two hairpins separated by a single-stranded H box with a consensus sequence 5′-ANANNA-3′, where N represents any nucleotide. The structure ends with a single-stranded 3′ tail that includes three ACA nucleotides [129]. Small nucleolar RNAs (snoRNAs) and small Cajal body-specific RNAs (scaRNAs) harbor an H/ACA motif. These families of small RNAs are mainly involved in the modification of ribosomal RNAs (by snoRNAs) and small nuclear RNAs (by scaRNAs), wherein the specific sites for the conversion of uridine to pseudouridine are identified [151]. To date, no telomerase RNA target for pseudouridylation has been identified. The distinguishing feature of hTR, as compared to other H/ACA RNAs, is the presence of P6.1 and P6b helices [152] located on the 5′ hairpin within the conservative CR 4/5 region. Additionally, hTR contains a BIO box on the 3′ hairpin, which is unique with respect to other human H/ACA RNAs. The BIO box assists in the assembly of the telomerase RNP [153]. The 3′ hairpin of hTR also includes a Cajal body box, which is referred to as a CAB box [154] (Figure 3A).

In yeast, H/ACA snoRNAs are transcribed as separate RNA molecules by RNA polymerase II and processed from synthesized precursors [155]. In ciliates, telomerase RNA is also expressed independently, but by RNA polymerase III [156]. Interestingly, in humans, H/ACA snoRNAs and scaRNAs are usually produced as processing products of spliced introns from mRNA [157]. Therefore, the biogenesis of telomerase RNA is distinct from the majority of human RNAs that carry the H/ACA motif. If hTR is transcribed by RNA polymerase III or if the H/ACA domain of hTR is transcribed within the intron of mRNA by RNA polymerase II, only the 3′ end of the RNA molecules carrying the H/ACA motif can be detected [129,134]. In contrast with other human H/ACA RNAs, processing in the 5′–3′ direction must be suppressed in order to preserve the pseudoknot and the template region.

The H/ACA motif is involved in the formation of a ribonucleoprotein complex with the proteins dyskerin, NHP2, NOP10, and GAR1 (H/ACA proteins) in all snoRNAs and scaRNAs of vertebrates [158,159]. Dyskerin, NHP2, and NOP10 were identified by telomerase immunoprecipitation followed by mass spectrometry [160]. Each H/ACA hairpin of telomerase RNA binds protein complexes consisting of dyskerin, NHP2, NOP10, and GAR1 [123]. Dyskerin, NHP2, and NOP10 have RNA-binding activity; GAR1 is attracted through protein–protein interactions only [124]. However, these proteins are not capable of the independent formation of a complex with an H/ACA motif in vivo without auxiliary proteins [161,162].

The assembly of the telomerase complex mediated by the H/ACA motif proceeds as follows. First, the assembly factor SHQ1 binds to dyskerin in the cytoplasm and stabilizes it, presumably preventing non-specific binding to RNA and non-specific pseudouridinylation [163,164]. The dyskerin–SHQ1 complex is then imported into the nucleus via the nuclear localization signal on the dyskerin. After its import into the nucleus, SHQ1 is separated from dyskerin by the R2TP chaperone complex consisting of the target-recognizing proteins PIH1D1, RPAP3, and the AAA+ ATPases RUVBL1 and RUVBL2 (also known as pontin and reptin) [162]. The assembly of the H/ACA complex may be facilitated by the NUFIP protein, which binds NHP2 and interacts with PIH1D1 [165]. The disruption of the activity of any of these assembly factors leads to accumulation disorders of the mature telomerase RNA in vivo [162,166]. The binding of dyskerin to telomerase RNA occurs during transcription in order to ensure the proper processing of telomerase RNA [167,168]. Chaperones place two tetramers [123], each of which consists of dyskerin, NHP2, NOP10, and the assembly factor NAF1; the latter is later replaced by structurally similar GAR1 [167,169] through an unknown mechanism speculated to involve the SMN protein [167]. Replacement occurs before the transportation of the H/ACA ribonucleoprotein into Cajal bodies or nucleoli, as NAF1 is only found in the nucleoplasm and thus not in Cajal bodies or nucleoli [161,167]. NAF1 may be recruited to the C-terminal domain of RNA polymerase II, thus promoting snoRNP assembly [170]. The mechanism of the tetramers’ assembly and R2TP complex recruitment remains unclear.

Experiments involving the mutagenesis of the 3′ and 5′ hairpins in H/ACA snoRNA and the cryo-electron microscopic (cryo-EM) structure of human telomerase indicate that the first dyskerin, NHP2, NOP10, and NAF1 tetramer binds to the 3′ hairpin during the assembly of the H/ACA ribonucleoprotein complex. This initial binding allows for the second tetramer to assemble on the 5′ hairpin [123,124]. The proteins dyskerin, NHP2, NOP10, and NAF1 are necessary for the stability of telomerase RNA and other H/ACA RNAs in vivo [171,172,173,174]. Despite the fact that GAR1 is a stoichiometric partner for the binding of snoRNAs with the H/ACA motif, it is not necessary for their stability in vivo [172]. GAR1 may be less associated with the telomerase complex compared to dyskerin, NHP2, and NOP10, according to the results of immunoprecipitation [160]. This is consistent with the model of NAF1 replacement by GAR1 at later stages of telomerase assembly.

The interaction between H/ACA tetramers (dyskerin/NHP2/NOP10/GAR1) in the structure of the human telomerase complex explains why mutations in the case of DC lead to a disruption in the maintenance of telomere length and not defects in the biogenesis of spliceosomal and ribosomal RNPs [175]. DC mutations disrupt the interaction between H/ACA tetramers, which leads to the incorrect assembly of the H/ACA ribonucleoprotein on the 5′ H/ACA hairpin of telomerase RNA. hTR has a shortened version of the 5′ hairpin compared to snoRNA and scaRNA, which leads to poor H/ACA tetramer binding due to RNA–protein interactions alone [124].

### 3.4. Posttranslational Modifications in Biogenesis of Telomerase RNP

Posttranslational modifications also affect the biogenesis of H/ACA ribonucleoproteins. It has been proposed that the poly-ADP-ribosylation (PARylation) of dyskerin and GAR1 affects their ability to bind to RNA and form a telomerase complex [176]. In addition, numerous studies have revealed that dyskerin, GAR1, NHP2, NAF1, and R2TP chaperone proteins (RPAP3, pontin, and reptin) undergo modifications with small ubiquitin-like modifiers (SUMO) [177,178,179,180,181]. The covalent posttranslational modification of SUMO, termed SUMOylation, is involved in a variety of processes in the cell [182]. Recent work has revealed several SUMOylation sites in the lysine-rich region of the nuclear/nucleolar localization signal at the C-terminus of dyskerin, the most important of which is K467. The replacement of K467R leads to a loss of localization of dyskerin in the nucleolus and a drop in telomerase activity in vitro [183]. In addition, GAR1 contains a hydrophobic motif that interacts with SUMO and promotes the effective binding of dyskerin to GAR1 [183]. It has been proposed that dyskerin dissolves in the dense fibrillar component of the nucleolus due to the motif in GAR1 interacting with SUMO, which recognizes this modification on dyskerin [183]. Interestingly, recent work has shown that one-third of the dyskerin molecules are statically associated with the nucleolus [184]. We speculate that this might be related to dyskerin SUMOylation. Surprisingly, the functioning of reptin and pontin is also regulated by SUMOylation [185,186]. The role of SUMOylation in the functioning of the assembly factors and components of the telomerase complex has yet to be properly assessed.

In addition to SUMOylation, the single-strand selective monofunctional uracil DNA glycosylase 1 (SMUG1) is involved in the nucleolar localization of dyskerin [187]. Mouse embryonic fibroblasts with SMUG1 homozygous knockout exhibit the mislocalization of dyskerin [188]. Quantitative phosphoproteomics has shown that H/ACA proteins, dyskerin, GAR1, NHP2, NOP10, and NAF1 change their phosphorylation status during the cell cycle [189], which may also affect telomerase biogenesis and its functioning.

### 3.5. Assembly of Active Telomerase Complex

The difference between scaRNAs and snoRNAs is the presence of a conservative sequence of four nucleotides, namely, a CAB-box necessary for localization in Cajal bodies [154]. Cajal bodies are dynamic and membraneless organelles found in the nucleus of eukaryotic cells. Cajal bodies are involved in the maturation and processing of ribonucleoproteins, including small nuclear RNPs (snRNPs) and small Cajal-body-specific RNPs (scaRNPs) [190]. Telomerase RNA contains sequences of H/ACA and CAB boxes and is localized in Cajal bodies, as well as scaRNAs, only after the processing and attachment of H/ACA proteins, according to the results of in situ hybridization [134,191,192].

The co-purification of dyskerin complexes from tumor cell lines allowed the detection of the protein TCAB1 (also known as WDR79 or WRAP53), which is associated with telomerase [193]. TCAB1 stably binds scaRNAs but not snoRNAs. This protein is localized in Cajal bodies but not in nucleoli. The first studies showed that TCAB1 knockdown leads to the localization of telomerase RNA outside Cajal bodies, presumably in nucleoli, and ineffective telomere elongation without a drop in telomerase activity in vitro [193,194]; recently, this finding was confirmed [184]. However, later studies have shown that TCAB1 knockout also leads to a drop in telomerase activity in vitro in cancer and embryonic stem cells without a consistent change in hTR accumulation [195,196]. TCAB1 is proposed to mediate the correct folding of the distant P6b and P6.1 loops of telomerase RNA, thereby enabling the effective interaction of the CR4/5 domain and hTERT and moderately stimulating telomerase activity in vitro [196]. Cryo-EM enabled the establishment of the fact that TCAB1 is a stable subunit of the telomerase holoenzyme [197]. Chaperone TRiC is required for TCAB1 folding and correct telomerase assembly [198]. TCAB1 is released from hTR in mitotic cells coincident with TCAB1 delocalization from Cajal bodies. At the same time, the total hTR level, the total TCAB1 protein level, and the telomerase activity in vitro remains consistent across the cell cycle, suggesting that TCAB1 may allow the telomerase holoenzyme to elongate telomeres [199].

hTERT interacts with the CR4/5 domain and the pseudoknot/template domain of hTR through the TRBD domain. In addition, TEN domains also shape the pseudoknot/template domain to stabilize the RNA–DNA duplex at the template’s 3′ end [200]. It has been proposed that the assembly of the hTERT–hTR complex occurs with the assistance of the chaperone Hsp90 [201,202], which is involved in the regulation of the cell cycle, the maintenance of the integrity of chromosomes, and other signaling pathways [203]. The p23 protein forms a complex with Hsp90. The suppression of p23’s functioning leads to the downregulation of telomerase activity in vitro [204,205]. The inhibition of Hsp90 by geldamycin lowers the content of the active telomerase complex and causes the degradation of hTERT in the proteasome. The immunoprecipitation of Hsp90 and p23 leads to the enrichment of active telomerase, which indicates their interaction with a mature telomerase complex [206]. The treatment of cells with geldanamycin also leads to a loss of NHP2 protein, indicating the involvement of Hsp90 in NHP2 stabilization [165]. However, the interpretation of the effect of Hsp90 is difficult, since the disruption of one of the key chaperones in the cell can have an indirect effect.

Another protein, AAA-ATPase NVL2, may also act as an hTERT chaperon. NVL2 interacts and co-localizes with hTERT in the nucleolus. NVL2 depletion decreases hTERT levels and telomerase activity in vitro [207]. In addition, it has been shown that the expression of the dominant-negative form of the snRNP assembly factor survival of motor neuron (SMN) disrupts the localization of hTERT in vivo and telomerase activity in vitro [208]. SMN may play a role in the assembly of a catalytically active telomerase complex [208] since SMN associates with GAR1 in vivo [209,210]. The SMN complex is concentrated in nuclear bodies, where it may promote NAF1–GAR1 exchange.

Interestingly, the localizations of hTR and hTERT only overlap at telomeres for most of the cell cycle [199]. Unlike hTR, hTERT tends to localize in parts of the nucleus other than Cajal bodies, especially in the nucleoli in cancer cells [191,211]. While the PinX1 protein may facilitate the nuclear localization of hTERT, this has only been observed in the context of overexpression [212]. However, a recent article has argued that endogenous, tagged hTERT is excluded from nucleoli [184].

There are currently several models of hTR and hTERT assembly. The first model suggests that this assembly occurs through the interaction of hTR-bearing Cajal bodies and hTERT-bearing nucleoli. As in the S phase of the cell cycle, the concentration of hTERT shifts from the nucleoplasm to the nucleoli [213,214], and Cajal bodies are assumed to move to the periphery of the nucleoli, carrying telomerase RNA with them [214]. The second model suggests that assembly occurs in nucleoli. It has been shown that hTERT can localize in the nucleoli and may bind to RNP in the nucleolar dense fibrillar component [215,216], while continuing to interact with nucleolin [216,217], until the mature complex is attracted to Cajal bodies by TCAB1 [216]. However, the active role of nucleoli in telomerase assembly is disputable [218]. The third model suggests that assembly occurs in Cajal bodies. According to structured illumination microscopy, hTR is located on the periphery of Cajal bodies. This suggests that hTERT may gather telomerase RNA at the exit from Cajal bodies [219]. In principle, the results obtained do not contradict the possibility that assembly may occur simply in the nucleoplasm without the participation of special nuclear compartments.

Despite numerous experiments that have been conducted to determine its localization, the exact location of hTERT–hTR assembly in the nucleus remains elusive. This can be partly explained by the limitations of the methods used. For example, it has been observed that the N-terminal tagging of hTERT affects its functioning in cells [220]; alternatively, the cause could be hTERT overexpression. Another important issue is the liquid–liquid phase separation during the formation of various nuclear bodies involved in telomerase biogenesis. This topic is discussed in further detail below.

Recent advances in cryo-electron microscopy have revealed that the H2A and H2B histone dimer is also a telomerase subunit [197]. Interestingly, H2A–H2B binds to the P6.1 stem in the catalytic lobe of telomerase and not in the H/ACA lobe [197]. The P6.1 stem in the CR4/5 domain is highly conserved among mammals [152]. Histones are also highly conserved proteins [221]. Therefore, it can be anticipated that the H2A–H2B dimer is part of the telomerase complex in other mammals, possibly contributing to the correct folding of the CR4/5 domain. Currently, there are no data confirming the stage at which the H2A–H2B dimer joins the telomerase complex. However, we can assume that this happens after hTERT–hTR assembly due to hTR folding. Further studies should reveal the role of this dimer with respect to the functioning of telomerase.

The hypermethylation of the telomerase RNA 5′ cap by trimethylguanosine synthase 1, TGS1, also plays an important role in telomerase trafficking and recruitment. Two differentially distributed isoforms of TGS1 have been found [222]. Besides Cajal bodies, the full-length isoform may localize in the cytoplasm, whereas the shorter isoform is located solely in Cajal bodies and associates with components of box C/D and H/ACA snoRNPs [222]. Interestingly, treatment with an inhibitor of TGS1, sinefungin, significantly reduced the number of Cajal bodies in cancer cells and tumor organoids [223]. Upon TGS1 knockout, the number of Cajal bodies also reduced, and scaRNAs tended to be mislocalized in nucleoli [131]. Meanwhile, hTR fluorescence in situ hybridization (FISH) combined with anti-TRF2 immunofluorescence revealed a significantly reduced degree of recruitment of telomerase RNA to chromosome ends upon TGS1 knockdown [223]. The removal of TGS1 also led to the accumulation of telomerase RNA in the cytoplasm. In addition, the total amount of hTR increased without changing the content of unprocessed forms of telomerase RNA [131]. Therefore, the TGS1-mediated hypermethylation of the hTR 5′ cap may, in principle, limit telomere elongation. Recent work proposes that the 2,2,7-trimethylguanosine capping of human telomerase RNA by TGS1 is required for direct telomerase-dependent telomere maintenance, although 2,2,7-TMG capping itself is dispensable with respect to telomerase activity in vitro [223].

### 3.6. Transport of Human Telomerase RNP

The trafficking of the telomerase complex is the most controversial aspect of its biogenesis. The main issue is how subnuclear compartmentalization affects the maturation of the hTR and telomerase complex. Currently, all works are focused on revealing the roles of Cajal bodies and nucleoli in this process.

FISH of 3′-extended telomerase RNA suggests that at least part of hTR 3′ end processing takes place in the nucleolus [224]. At the same time, the interaction of hTR with TCAB1 leads to the concentration of telomerase RNPs in Cajal bodies [193,194,219]. A loss of TCAB1 leads to the nucleolar accumulation of hTR [184]. A notable finding from crosslinking studies of the interactions between coilin, the main component of Cajal bodies, and RNA is that all snoRNAs may migrate through Cajal bodies to the nucleolus, while scaRNAs are uniquely retained therein [225].

The transport of scaRNAs into Cajal bodies is carried out by the PHAX factor. Presumably, m^7^G-capped telomerase RNA is also transported by PHAX since it is associated with hTR [226]. Nopp140, an intrinsically disordered Cajal body phosphoprotein, co-purifies with dyskerin, as does TCAB1 [193]. Nopp140 is required to recruit and retain all scaRNPs in Cajal bodies. Nopp140 plays an important role in the formation of Cajal bodies and the localization of scaRNAs and the telomerase RNA within them [227]. Nopp140 may also function as a transport factor between the nucleolus and Cajal bodies [228].

The trafficking of hTR to telomeres and Cajal bodies also depends on hTERT in cancer cells. The depletion of hTERT leads to a loss of hTR from both Cajal bodies and telomeres without a change in hTR levels [229]. At the same time, hTERT overexpression also leads to a reduction in the localization of MS2-tagged hTR in Cajal bodies [219]. Live-cell analysis of the diffusion coefficients of tagged hTERT revealed three separate populations of telomerase particles. There were two rapidly diffusing populations, which may represent unbound hTERT and diffusing telomerase RNPs, and a less mobile population that may represent telomerase RNPs bound to Cajal bodies or telomeres [230]. Interestingly, even in cells without hTR, 25–30% of hTERT particles were slowly diffusing or static [184]. All these results suggest that there could be structures other than Cajal bodies, nucleoli, or telomeres that retain hTERT to achieve optimal telomerase assembly. It is possible that the proline/arginine/glycine-rich region of hTERT is needed for this to occur. Retainment can be performed by other nuclear bodies, for example, nuclear speckles.

Therefore, we would like to note that the current approach to considering the role of Cajal bodies and nucleoli in telomerase biogenesis has a number of limitations. Cajal bodies and nucleoli are not entirely separate structures but have complex relationships. They can sometimes overlap, as evidenced by electron microscopy and the co-localization of Cajal-body-related and nucleolar factors [231]. Cajal bodies need nucleoli to maintain their integrity, but not vice versa. This indicates that Cajal bodies may have an auxiliary function. In principle, the nucleoli can perform at least part of Cajal bodies′ functions, depending on the conditions. In our opinion, this explains a number of contradictions in the literature regarding their role in telomerase biogenesis. Additionally, in principle, other nuclear bodies might participate in human telomerase biogenesis in a less pronounced way.

### 3.7. hTR Modifications in Telomerase Biogenesis

Post-transcriptional modifications may affect the functioning of telomerase RNA. Cytosines C106, C166, C323, and C455 have been identified as m5C sites in hTR [232,233]. It was shown that the RNA-binding protein HuR associates with hTR and promotes the methylation of C106 by an unknown methyltransferase. This modification can change the secondary structure of hTR, thus affecting the association of hTERT and hTR [233]. A subsequent study revealed that neural-specific HuB and HuD compete with HuR during hTR binding and antagonize HuR’s functions [234].

An initial study of hTR pseudouridinylation sites identified several candidate bases, including U159, U161, U179, U306, U307, U316, and U370. The pseudouridylation of U306 and U307 changes the conformation of the highly conserved P6.1 hairpin [235]. A large-scale search confirmed the occurrence of the significant pseudouridinylation of U307, as well as a less pronounced modification of U179 [236]. Later, a chemical-probing analysis helped to identify 18 pseudouridines in hTR [237]. It can be assumed that the pseudouridinylation of U307 may play a role in the correct folding of the CR4/5 domain and in its interaction with the histone dimer H2A–H2B.

Telomerase RNA may carry other post-transcriptional modifications that affect the processing, transport, and assembly of the telomerase complex. There is indirect evidence that hTR has several modified bases between the CR4/CR5 domain and H box [188]. Currently, the functional role of hTR modifications remains poorly understood.

### 3.8. hTERT Biosynthesis Regulation

hTERT is a limiting factor in the formation of an active telomerase complex. The estimated half-life of telomerase activity in vitro is no more than 24 h [238,239], which is less than 5 days of telomerase RNA’s half-life [150]. At the same time, the level of hTR remains constant during cell cycle progression [199,216]. Based on this, it can be assumed that the stability control of telomerase protein components, including hTERT, contributes to the regulation of telomerase. The expression of *hTERT* is actively regulated by different transcriptional factors and epigenetic modifications (reviewed in [240]). The *hTERT* promotor is often mutated in different types of cancers [241].

#### 3.8.1. hTERT Alternative Splicing

The *hTERT* gene consists of 16 exons and 15 introns. The full-length isoform of the 16 exons is only capable of elongating telomeres [242]. Twenty-two isoforms of hTERT mRNA have been identified [243]. The most studied alternative splice variants of hTERT encode proteins lacking catalytically active RT and are generated by the alternative splicing of the α and/or β sites. Skipping 36 nucleotides results in an α– isoform, whereas the β– isoform produced by 183-nucleotide deletion (exons 7 and 8) harbors a premature termination codon. The β– protein competes with full-length hTERT in binding to hTR, thereby inhibiting telomerase activity in vivo [242]. The same mechanism was proposed for the α– isoform [244].

Splicing may be directed by the action of RNA-binding proteins. The production of full-length hTERT transcripts is promoted by NOVA1, which enhances the inclusion of exons in the RT domain of hTERT [245]. A recent study proposed that the developmental control of telomerase activity in vivo is driven by the alternative splicing of *hTERT* exon 2. Protein SON promotes the skipping of exon 2, which triggers hTERT mRNA decay in differentiated cells [246].

#### 3.8.2. hTERT Localization

hTERT must move from the cytoplasm to the nucleus after its synthesis to produce an active telomerase complex. The hTERT nuclear localization signal (NLS) includes two clusters of basic amino acids [247]. The nuclear localization of hTERT requires the classic nuclear import machinery involving importins α/β and Ran GTPase. Importin α binds to hTERT N-terminal’s nuclear localization signal, while its partner importin β1 interacts with a nuclear pore complex [248]. The Hsp90–FKBP52 complex also mediates hTERT nuclear import. Hsp90-binding immunophilins, FKBP51 and FKBP52 (also known as FKBP5 and FKBP4), engage in co-immunoprecipitation with hTERT [249]. The FKBP52 co-chaperone interacts with the hTERT–Hsp90 complex and promotes the nuclear transport of hTERT via a dynein/dynactin-dependent mechanism. The depletion of FKBP52 results in the cytoplasmic accumulation of hTERT and its ubiquitin-dependent proteolysis [250].

Interestingly, hTERT contains potential signals of nuclear import and export [251]. This indicates that the shuttling of hTERT between the nucleus and the cytoplasm may be one of the forms of telomerase regulation [247].

#### 3.8.3. hTERT Phosphorylation

A number of works have revealed that hTERT is phosphorylated at different sites. Five putative phosphorylation sites have been reported: serine 227 [247], threonine 249 [252], serine 457 [253], tyrosine 707 [254], and serine 824 [255].

The phosphorylation of serine 227 is required for hTERT’s translocation in the nucleus [247,248]. The Akt-mediated phosphorylation of S227 increases hTERT′s affinity for importin-α and promotes the nuclear import of hTERT [248].

A recent study identified that hTERT is phosphorylated at threonine 249 during mitosis by the serine/threonine kinase CDK1. The phosphorylation of threonine 249 is necessary for hTERT-mediated RNA-dependent RNA polymerase activity but is not required for reverse transcriptase activity in vitro [252]. An analysis of clinical samples revealed that the phosphorylation of threonine 249 is associated with aggressive phenotypes in various types of cancer [256].

Dual-specificity tyrosine-(Y)-phosphorylation-Regulated Kinase 2 (Dyrk2) phosphorylates serine 457 of hTERT. The phosphorylated hTERT associates with the EDD–DDB1–VprBP E3 ligase complex for subsequent ubiquitin-mediated hTERT protein degradation. Dyrk2 interacts with hTERT during the G2/M phases [253], which could be the mechanism for the cell-cycle-dependent regulation of telomerase activity in vivo.

Src kinase has been shown to regulate the nuclear export of hTERT under oxidative stress by phosphorylating tyrosine 707 [254,257]. In turn, protein tyrosine phosphatase Shp-2 counteracts Src kinase. Shp-2 promotes the retainment of hTERT in the nucleus via the downregulation of tyrosine 707’s phosphorylation [257].

It has been shown that the phosphorylation of hTERT serine 824 by Akt kinase and kinase C correlates with increased telomerase activity in vitro [255,258]. This is presumably due to the enhanced translocation of hTERT into the nucleus from the cytoplasm [255,258]. At the same time, hTERT modification by c-Abl kinase leads to a threefold decrease in telomerase activity in vitro. The knockout of c-Abl in a mouse model lead to increased telomerase activity in vitro and better telomere elongation [259].

Serine/threonine-protein phosphatase 2A (PP2A) is proposed to dephosphorylate hTERT and inhibit telomerase activity in vitro by accumulating hTERT in the cytoplasm [260].

#### 3.8.4. hTERT Ubiquitination and SUMOylation

The ubiquitination of hTERT plays a role in the regulation of telomerase activity in vivo by regulating its stability. It has been shown that E3-ubiquitin ligase MKRN1 interacts with hTERT in a yeast two-hybrid system. MKRN1 overexpression leads to the degradation of hTERT [261]. In addition, the effect of MKRN1 on telomerase activity during cell differentiation has been shown. The human leukemia cell line HL-60 expresses MKRN1 at a low level. However, with the induction of differentiation, expression increases significantly, which is combined with a significant decrease in telomerase activity in vitro [262]. Thus, the degradation of hTERT mediated by MKRN1 can provide a decrease in telomerase activity in vivo in differentiated cells when it becomes unnecessary. Another interesting observation is that lysophospholipid sphingosine 1-phosphate (S1P), which is generated by sphingosine kinase 2 (SK2), binds hTERT at the nuclear periphery in human and mouse fibroblasts. S1P binding inhibited the interaction of hTERT with MKRN1 [263].

E3 ubiquitin ligase MDM2 (HDM2) can interact with hTERT through multiple domains on both proteins. In this case, hTERT undergoes polyubiquitination and degradation by the proteasome. The removal of MDM2 leads to increased hTERT content in cells and increased telomerase activity in vitro [264]. Interestingly, the E2-ubiquitin-conjugating enzyme UBE2D3, which is one of the partners of MDM2, can also affect the ubiquitination of hTERT. The overexpression of UBE2D3 also leads to a lower level of hTERT protein and decreased telomerase activity in vitro and in vivo [265].

In another study, it was shown that the CHIP E3 ligase regulates the stability of hTERT in the cytoplasm. Interaction with CHIP leads to polyubiquitination, prevents the transfer of hTERT into the nucleus, and completes the proteolytic degradation of hTERT [266]. This interaction peaks during G2/M phases and decreases during S phase. At the same time, telomere elongation occurs. Thus, CHIP might modulate telomerase activity in vivo throughout the cell cycle by controlling the trafficking and stability of hTERT [266].

It has been proposed that the Plk1 protein is associated with telomerase and promotes the retention of hTERT in the nucleus, preventing its ubiquitination and degradation in the cytoplasm [267].

A recent study reported that hTERT is SUMOylated by SUMO1 at lysine 710. The polycomb protein CBX4 acts as the SUMO E3 ligase of hTERT. The SUMOylation of hTERT results in the upregulation of telomerase activity in vivo, which can be inhibited by the process of SENP3-mediated deSUMOylation. Interestingly, it has been discovered that hTERT SUMOylation plays a role in the repression of E-cadherin gene expression. This can lead to the activation of the epithelial–mesenchymal transition (EMT) in breast cancer cells [268]. There are many other examples of hTERT functions besides telomerase in the literature, including the regulation of gene expression or mitochondrial function and oxidative stress response (reviewed recently in [269]).

#### 3.8.5. hTERT Sequestration

Negative regulation of telomerase activity may occur due to the arrest of telomerase complex in the nucleolus when hTERT temporarily moves from the nucleoplasm to the nucleolus. It is assumed that this translocation reduces the likelihood of telomere formation at the ends of damaged DNA [213]. PIN2/TRF1-interacting telomerase inhibitor 1 (PinX1) can regulate telomerase activity by arresting hTERT. PinX1 has been shown to bind directly to the telomere protein TRF1 [270] as well as hTERT and telomerase RNA and inhibits telomerase activity in vitro [271] and in vivo upon overexpression [272,273]. However, the silencing of PinX1 leads to telomere shortening in telomerase-positive cancer cells of various origins [274,275], indicating its dual role in telomere length maintenance. Nucleophosmin (NPM) can partially attenuate the PinX1 inhibition of telomerase activity in vitro, and NPM loading to hTERT requires PinX1 [276]. hTERT/PinX1/NPM interaction peaks during telomere extension in S phase [277]. Human microspherule protein 2 (MCRS2) is also a negative regulator of telomerase activity in vitro and in vivo. MCRS2 binds to PinX1 and is colocalized with it in the nucleus and on telomeres. At the same time, its expression is limited to S phase, unlike PinX1. MCRS2 could be another partner of PinX1, which regulates its operation in S phase [278]. Despite active research, the exact functional role of the PinX1 protein remains elusive.

### 3.9. Delivery of Telomerase Holoenzyme to Telomeres in Vertebrates

hTERT expression is necessary for hTR localization in telomeres. Human telomerase mainly associates with telomeres solely during S phase [167,183,231]. Most enzymes meet the substrate by simple diffusion. However, it is estimated that telomerase and its telomere substrate have very low concentrations in a normal cell: approximately 250 telomerase holoenzymes per 184 telomeres during late S phase [252]. Therefore, a special mechanism may be required to recruit telomerase to telomeres. However, we note that the accurate mathematical modeling of the process of attracting telomerase to telomeres and their elongation is required. Therefore, low telomerase content does not indicate the presence of a special mechanism for the delivery of telomerase to telomeres.

Initial in situ hybridization experiments demonstrated that Cajal bodies are in contact with some telomeres. These results led to the hypothesis that Cajal bodies move through the nucleoplasm and deliver telomerase RNA to telomeres in the S phase of the cell cycle [214,279]. The following studies showed that the loss of coilin, the structural protein of Cajal bodies, disrupts Cajal body formation and telomerase recruitment to telomeres [280,281]. A CAB box mutant of hTR fails to accumulate in Cajal bodies and forms an active telomerase complex, but is strongly inefficient with respect to telomere extension [282]. The overexpression of hTR and hTERT leads to the formation of new Cajal bodies on telomeres according to one study’s FISH results [281].

However, the results of several subsequent studies do not support the model wherein the delivery of the telomerase holoenzyme to telomeres is performed exclusively by Cajal bodies. First, it was shown that a minimal portion of telomerase RNA can be assembled without the H/ACA motif, and this portion can form an active telomerase that effectively elongates telomeres, thereby bypassing the assembly pathway through the H/ACA motif and Cajal bodies [195]. In addition, localization in Cajal bodies is not strictly necessary to maintain telomere length in all cells since the removal of coilin does not lead to the impairment of telomere elongation in cancer cells [195,283] but, instead, a modest increase in telomerase activity in vitro [196]. TCAB1/hTR foci are detected transiently during S phase at telomeres in the absence of coilin [283]. Even the earlier study supported the notion that TCAB1 can localize to telomeres in Cajal body-independent manner [280]. Surprisingly, the absence of telomerase in Cajal bodies due to the knockdown of Nopp140 leads to gradual telomere elongation in cancer cells [227]. Moreover, recent imaging data obtained from living cells have shown that no more than 10% of hTR is localized in Cajal bodies; the rest is distributed throughout the nucleoplasm [219]. Meanwhile, hTERT overexpression does not enhance the co-localization of the telomeric protein TRF1 with the Cajal body protein coilin. Despite this, hTR resides in Cajal bodies for a longer time than in normal diffusion [219]. Interestingly, telomerase trafficking in mice is reported to be Cajal-body-independent [284]. Numerous studies have demonstrated that human telomerase resides in Cajal bodies for a considerable proportion of its life cycle. However, the functional significance of this remains unclear. Perhaps the role of Cajal bodies is more pronounced with low telomerase expression in normal cells. This remains to be clarified, as most studies have been performed using cancer cell lines or in the context of telomerase overexpression. We speculate that the telomerase complex may diffuse between the nucleoli, Cajal bodies, and nucleoplasm. However, in normal conditions, the presence of TCAB1 shifts the balance in favor of Cajal bodies. Thus, Cajal bodies limit telomerase activity in vivo rather than promoting it by sheltering the assembled telomerase complex.

### 3.10. Telomerase Recruitment to Telomeres in Vertebrates

In vertebrates, telomeres are protected by a special protein complex (shelterin), which consists of six proteins found in humans: TRF1, TRF2, TIN2, RAP1, POT1, and TPP1 [285,286] (Figure 4). Shelterin prevents the recognition of the ends of chromosomes as double-stranded DNA breaks [287]. TRF1 and TRF2 proteins directly bind double-stranded DNA as homodimers [288,289], while POT1 protein binds single-stranded DNA [290,291]. TPP1 interacts with both TIN2 and POT1 and promotes POT1–telomere binding [292]. TIN2 interacts with both TRF1 and TRF2 and tethers shelterin [286]. RAP1 binds to TRF2 and contributes to telomere protection [293]. To learn more about shelterin functioning, refer to a recent review [294].

The recruitment of human telomerase to telomeres requires the proteins TIN2 and TPP1 [281,295]. TPP1 interacts directly with the TEN domain of hTERT via a patch of amino acids known as the TEL patch [281,296]. In turn, TIN2 performs a critical bridging function that is necessary for telomerase recruitment [295]. It has been shown that the POT1–TPP1 complex enhances the processivity of the minimal telomerase complex in vitro by maintaining the association of the complex with the DNA product and during its translocation [297]. In addition, the POT1–TPP1 complex prevents the binding of the RPA protein to the 3′ telomere overhangs, thereby limiting the activation of the DNA damage response [219].

Notably, the CTC1/STN1/TEN1 complex, which comprises human orthologues of the yeast CST complex, can bind to the 3′ single-stranded telomere overhang and obstruct telomerase’s access to the telomere during the late S/G2 phases [298,299]. Recent cryo-EM results showed that CST forms a ring-like decameric DNA–protein supercomplex [300].

The single-stranded telomere overhang can invade the double-stranded telomere region to form a displacement loop (D-loop) and a telomere loop (T-loop). The T-loop’s formation is regulated by shelterin, and TRF2 plays a crucial role in this process [301]. T-loop restricts telomerase’s access to the single-stranded overhang [302,303].

RTEL1 helicase is a crucial component of telomere homeostasis, and mutations in RTEL1 cause a severe form of DC known as Hoyeraal–Hreidarsson syndrome [304], which is a hereditary disorder associated with severely shortened telomeres and diverse clinical symptoms. RTEL1 is recruited to shelterin by TRF1 and TRF2 [305,306,307]. RTEL1 is essential for the proper disassembly of T-loop and the unwinding of telomere G-quadruplexes during S phase [307,308,309]. RTEL1 was also shown to stabilize long, single-stranded telomere overhangs [310].

hTERT recruitment to telomeres during the S phase is dependent on the dissociation of TRF1 from the telomere, a process promoted by ATR/ATM kinases [311,312]. Phosphorylated TRF1 is subsequently directed to proteasomal degradation [313]. The dissociation of TRF1 releases the 3′ telomeric overhang from a protective T-loop with the assistance of RTEL1, allowing for telomere elongation via telomerase [314]. ATM was shown to collaborate with the MRE11–RAD50–NBS1 (MRN) complex to promote telomere elongation by the 5′ end processing of telomeres, thus resembling the yeast mechanism [315,316].

In summary, the recruitment of vertebrate telomerase to telomeres is also regulated by post-translational modifications of telomeric proteins. While there are other modifications of telomeric proteins, their addressal is beyond the scope of this review. Some of these modifications were reviewed in [105,317,318].

In Figure 4, we combine the current data and propose a model of human telomerase biogenesis.
Figure 4Model of human telomerase complex and telomerase RNA biogenesis. (**A**) After its synthesis, dyskerin (DKC1) is stabilized by SHQ1 protein in the cytoplasm and transported into the nucleus. (**B**) Telomerase RNA (hTR) is expressed as a 3′ poly(A) precursor monomethylated at the 5′ cap (5′MMG) by RNA polymerase II (RNAP II) under the control of different transcriptional factors and influenced by epigenetic modifications of hTR promotor. (**C**) Upon synthesis, the first H/ACA tetramer (DKC1, NOP10, NHP2, and NAF1) is attached to 3′ H/ACA hairpin with the assistance of R2TP chaperon and NUFIP factor, which interact with NHP2. Subsequently, SHQ1 is released. (**D**) The second tetramer is assembled in the same fashion. (**E**) If the hTR precursor is long, then it is degraded through NEXT- or PAXT-dependent recruitment of the exosome [133]. (**F**) During the transport of the pre-assembled complex to the nucleolus facilitated by PHAX, the exchange of NAF1 for GAR1 may occur, possibly with the assistance of SMN protein. (**G**) PARN and RRP6 deadenylate hTR precursors in the nucleolus. The TRAMP complex polyadenylates hTR precursors and promotes their degradation by the exosome [133,224,319] (**H**) hTERT mRNA is produced by RNAP II. hTERT promotor is bound by many transcriptional factors, epigenetically modified, and often mutated in cancers. (**I**) hTERT mRNA can be spliced into different isoforms that compete with a full-length isoform. The full-length isoform is translated into a full-length hTERT protein that is stabilized by Hsp90–p23 chaperon in the cytoplasm. (**J**) Levels of active hTERT are controlled by ubiquitylation via MKRN1, CHIP, and MDM2 E3-ubiquitin ligases, as well as phosphorylation. (**K**) Importin complex transports hTERT to the nucleus through the nuclear pores. FKBP52 co-chaperone interacts with hTERT-Hsp90 complex and promotes hTERT nuclear transport via a dynein/dynactin-dependent mechanism. (**L**) After hTR’s processing, H/ACA pre-assembled complex is transported to Cajal bodies, probably by PHAX and Nopp140. hTERT and then H2A–H2B dimer join the RNP, likely at the periphery of Cajal bodies. 2,2,7-trimethylguanosine cap (5′TMG) is formed by TGS1 in Cajal bodies. Unfinished hTR processing may be completed in Cajal bodies. (**M**) hTR might be processed outside the nucleolus by another mechanism and delivered to Cajal bodies, as proposed in [320]. (**N**) Following translation, TCAB1 is stabilized by TRiC chaperonin and imported into the nucleus, where TCAB1 completes the telomerase holoenzyme. (**O**) Telomeres are protected by protein complex known as shelterin. The double-stranded telomere region is bound by TRF1 and TRF2 homodimers. TIN2 performs bridging function by binding TRF1 and TRF2. TIN2 also associates with TPP1. POT1 protein binds to single-stranded telomere region and cooperates with TPP1. RAP1 binds to TRF2. During S phase, 3′ single-stranded telomere overhang is released from the T-loop, allowing telomerase to access telomeres (not shown, discussed in Section 3.10) (**P**) Telomerase complex may be delivered to telomeres by Cajal bodies or independently. The interaction between TEL patch of TPP1 and TEN domain of hTERTenables the recruitment of telomerase to telomeres. PinX1 protein interacts with TRF1 and hTERT and inhibits telomerase recruitment to telomeres. Telomerase complex might diffuse between the nucleus, Cajal bodies, and nucleoplasm, but under normal conditions, TCAB1 shifts the balance toward Cajal bodies. (**Q**) hTR is exported to cytoplasm, possibly involving polyadenylate-binding nuclear protein 1 (PABPN1). It is not clear what happens to the proteins associated with the H/ACA motif. (**R**) hTR is the template for hTERP protein biosynthesis [321]. (**S**) Cytoplasmic hTR is imported into the mitochondrial intermembrane space (IMS) by PNPT1, where it is processed by ribonuclease T2 (RNASET2), leaving almost the entire pseudoknot domain. (**T**) The 195-nucleotide-long processed hTR (TERC-53) is exported into cytoplasm by unknown mechanism [138].
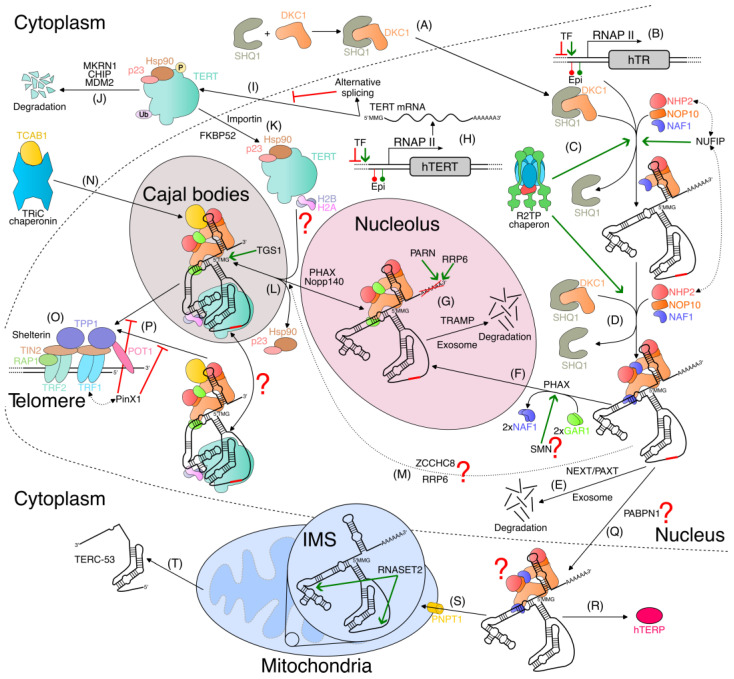



## 4. Common and Specific Motifs of Telomerase Biogenesis in Yeast and Vertebrates

Telomerase biogenesis in both yeast and vertebrates involves similar stages and molecular agents, but there are notable differences. In vertebrates, the regulation of telomerase biogenesis is more complex and is accurately tuned by various signaling pathways and chromatin modifications [240]. In both yeast and vertebrates, the biogenesis of the telomerase complex involves transcription by RNAP II and the processing of the telomerase RNA [44,45,133]. Prior to being bound by telomerase reverse transcriptase, yeast and vertebrate telomerase RNAs recruit proteins that remain part of the active holoenzyme and are required for telomerase RNA stability in vivo. While yeast utilizes Sm or Lsm proteins for this purpose [55,59], vertebrates employ H/ACA proteins [167,168]. Yeast and vertebrate telomerase RNAs are first capped by MMG at the 5′ end and later converted into the TMG cap in the later stages of biogenesis [22].

In humans, hTERT isoforms exist due to alternative splicing that increases regulatory complexity [243]. Hsp90 and p23 chaperones associate with human TERT and promote telomerase assembly [202,205]. Yeast Hsp90 and p23 orthologues have been shown to promote telomerase activity in vitro and in vivo by increasing the DNA binding of telomerase [202,321,322]. Their functional role in telomerase assembly is yet to be assessed. Hsp90 and p23 orthologues likely have a direct influence on telomere extension by telomerase in yeast, while in vertebrates, these chaperones mainly specialize in telomerase assembly.

Interestingly, both yeast and vertebrates control the assembly of the telomerase complex through compartmentalization, but in different ways. Yeasts most likely rely on the export of telomerase RNA to the cytoplasm and its subsequent assembly therein [57]. In vertebrates, nuclear bodies most likely play a decisive role [214,215,219].

The activity of the telomerase complex seems to be regulated more directly in yeast than in vertebrates by the process of assembly–disassembly. In yeast, Est1 is degraded in the G1 phase [106,107,108], and Est2 dissociates in G2/M phases [37], limiting telomerase assembly. Alternatively, telomerase activity in vitro does not change significantly throughout the cell cycle in humans. However, TCAB1 leaves the complex in the M phase [199], which should lead to a drop in telomerase activity in vitro [195,196]. In general, the exact mechanism behind the assembly–disassembly of the telomerase complex has not been well established in vertebrates.

Furthermore, the mechanisms of telomerase recruitment to the telomeres also differ, especially between budding yeast and vertebrates. In budding yeast, telomerase is recruited to telomeres through two distinct mechanisms: the Sir4–yKu80 pathway and the main Cdc13–Est1 pathway [86,87]. In vertebrates, the recruitment of telomerase to telomeres is mediated by multiple factors, including TPP1, POT1, and TIN2 [281,295]. The holoenzyme recruitment mechanisms also exhibit some similarities across the phylogenetic groups. The regulatory activities of Cdc13, Est1, and Est3 in yeast could parallel the roles of TIN2-bound TPP1 in vertebrates. The same considerations apply to the CST complexes.

The crucial stages of telomerase biogenesis and recruitment in yeast and vertebrates share some functional similarities. At the same time, the composition of the telomerase complex, the peculiarities of biogenesis regulation, and the mechanisms of recruitment to telomeres differ significantly between the two systems.

## 5. Prospects

It is known that the formation of membrane-free organelles such as Cajal bodies or nucleoli occurs due to liquid–liquid phase separation [323]. An interesting example is the nuclear bodies of promyelocytic leukemia involved in alternative telomere elongation (ALT) due to the clustering of telomere and DNA repair factors [324]. The condensation of these bodies occurs during the SUMOylation of telomeric proteins due to the interaction of SUMO and a motif interacting with SUMO (SIM) [325]. The further study of membrane-free organelles may reveal new details regarding the transport of telomerase within the nucleus and its recruitment to telomeres in yeast and vertebrates.

The limitation of telomerase activity in vivo by the end of the S phase in vertebrates remains poorly understood because the cellular level of active telomerase complex is not cell-cycle-regulated [199]. Interestingly, the level of interaction between the chaperones pontin and reptin with hTERT peaks in the S phase of the cell cycle [166]. These observations support the hypothesis of the cell-cycle-regulated activation of the telomerase complex in vivo. It is possible that telomerase regulation involves post-translational modifications of telomerase protein subunits and chaperones specific to the phases of the cell cycle in both yeast and vertebrates.

In addition, it is interesting to study the role of telomeric protein modifications in telomerase recruitment. For example, the level of TRF1 decreases due to poly-ADP-ribosylation by tankyrase, which leads to a weakening of the binding of TRF1 to telomeres [326]. ATM and ATR kinases promote telomeric elongation in human cells by increasing the frequency of the recruitment of telomerase to the telomere ends [219,311,312]. The main question is how the post-translational modifications of telomerase-associated and telomeric proteins achieve the timely adjustment of the assembly of the telomerase complex and its recruitment to telomeres. Thus, the study of these post-translational modifications and the modifications of telomerase RNA may help reveal new aspects of telomerase biogenesis and functioning.

In addition to investigating the biogenesis of the telomerase complex, studying the functions of hTERT and hTR beyond the context of telomerase is also an intriguing area of research. Although many of the observed effects are controversial, future studies should elucidate the non-canonical functions of hTERT and hTR.

## 6. Conclusions

The biogenesis of telomerase components, complex assembly, localization to the place of action, and recruitment to the telomeres comprise a complicated system, in which each step must be regulated to meet cellular requirements. Any impairment in the function or localization of the participants of telomerase biogenesis will affect the maintenance of telomere length, which is critical to processes such as regeneration, immune response, embryonic development, and cancer progression. At present, the key components of the telomerase complex in various organisms have been identified. In addition, great progress has been made concerning the determination of the structure of the telomerase complex. However, the functional roles of the post-translational modifications of telomerase-associated proteins and the modifications of telomerase RNA remains largely unexplored. Moreover, we still must elucidate the physical principles of the operation of nuclear bodies in general and their role in the biogenesis of the telomerase complex in particular.

## Figures and Tables

**Figure 1 ijms-24-05027-f001:**
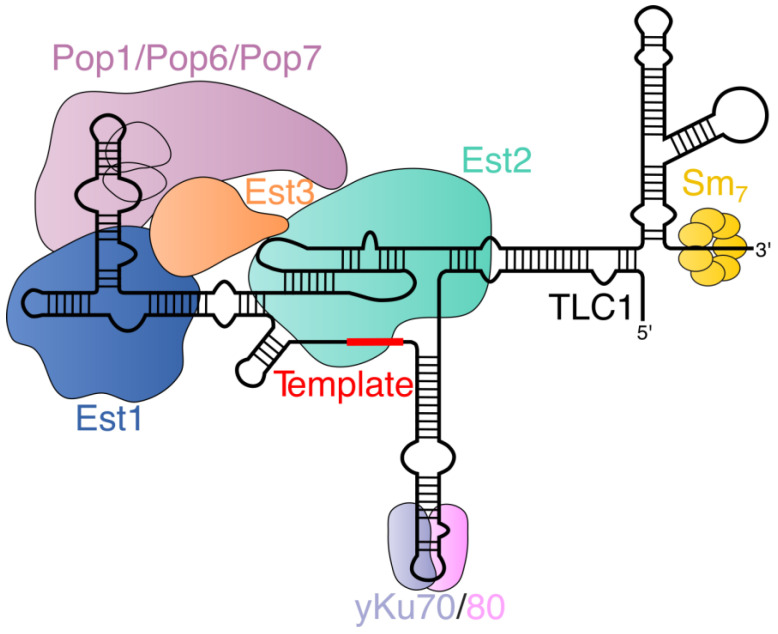
Scheme of telomerase holoenzyme’s composition in budding yeast. The secondary structures of TLC1 and protein components and the interactions between them are depicted in a simplified way, but in a manner consistent with current data. TLC1 works as a flexible scaffold for associated proteins. Ever shorter telomere (Est) proteins were first identified as important for telomere homeostasis. Est2 binds the pseudoknot (not marked) and template region of TLC1. Est1 and Pop proteins bind the same TLC1 arm. Pop proteins facilitate binding of Est1 and Est2 (and Est3 probably). Est3 does not interact with TLC1 and associates through Est1 or/and Est2 depending on the type of yeast. yKu70/80 heterodimer binds another TLC1 arm. Heptameric Sm_7_ protein ring protects the 3′ end of TLC1.

**Figure 2 ijms-24-05027-f002:**
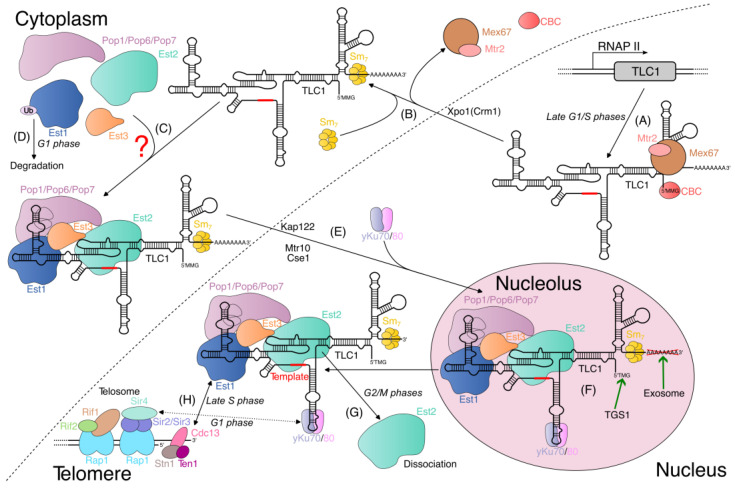
Model of telomerase biogenesis in budding yeast. (**A**) In late G1/S phases, telomerase RNA (TLC1) is synthesized by RNA polymerase II (RNAP II). Immature TLC1 has monomethylated 5′ cap (5′MMG) bound by cap-binding complex (CBC) and 3′ poly(A) tail (if it is not processed by exonucleolytic cleavage). Upon transcription, export receptor heterodimer Mex67-Mtr2 binds TLC1, stabilizing it and promoting its nuclear export. (**B**) Exportin Xpo1 ensures TLC1 export to cytoplasm, where Mex67-Mtr2 and CBC dissociate from it, while Sm_7_ complex binds 3′ end of TLC1. (**C**) Pop proteins, Est1, Est2, and Est3 are attached to the complex in the cytoplasm in an unknown order. (**D**) During G1 phase, Est1 protein is downregulated by ubiquitylation and subsequent degradation. (**E**) Assembled telomerase complex is imported to nucleus by the action of importin Mtr10 and karyopherins Kap122 and Cse1. After import, yKu70/80 binds telomerase RNP, possibly promoting its nuclear retention. (**F**) Telomerase complex moves to nucleolus, where poly(A) tail is removed and 5′-2,2,7-trimethylguanosine cap (TMG) is formed. (**H**) Budding yeast telomere is protected by a protein complex (telosome). The double-stranded region of the telomere is bound by Rap1 and associated Rif1/Rif2 proteins or Sir2/Sir3/Sir4 proteins. The single-stranded region is protected by Cdc13 in a complex with Stn1 and Ten1 proteins. During late S phase, yeast telomerase is recruited to telomeres by interaction between Cdc13 and Est1, which is supported by Est1 upregulation in S phase. yKu70/80 heterodimer can bind telomere through Sir4 protein in G1 phase, but this interaction does not promote telomere elongation. (**G**) In G2/M phases, Est2 dissociates from telomerase RNP, thereby limiting its activity during cell cycle progression.

**Figure 3 ijms-24-05027-f003:**
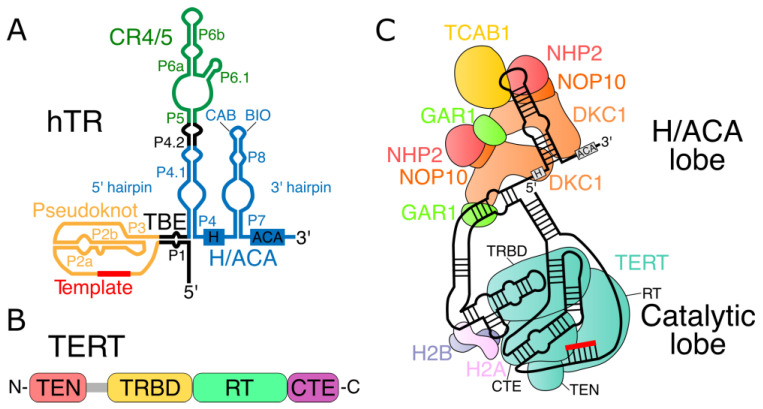
Scheme of hTR and hTERT structures and human telomerase holoenzyme. (**A**) Secondary structure of human telomerase RNA (hTR). CR4/5—conserved regions 4 and 5; TBE—template boundary element; H/ACA—H and ACA boxes; CAB—Cajal body box; BIO—biogenesis-promoting box. Pseudoknot, template, 5′ and 3′ hairpins of H/ACA domain, and P1, P2a, P2b, P3, P4, P4.1, P4.2, P5, P6a, P6b, P6.1, P7, and P8 elements are labeled. (**B**) Domain structure of telomerase reverse transcriptase (TERT) from N- to C-end. TEN—telomerase essential N-terminal domain; TRBD—telomerase RNA-binding domain; RT—reverse transcriptase domain; CTE—the C-terminal extension domain. (**C**) Scheme of human telomerase holoenzyme showing the relationships of hTERT domains, associated proteins, and hTR, adapted from the cryo-electron microscopy structure of human telomerase. Telomerase adopts RNA-tethered bilobed structure. The first lobe consists of H/ACA domain of hTR, which binds two tetramers (DKC1, NOP10, NHP2, and GAR1) and TCAB1. The second lobe is composed of hTERT as well as pseudoknot and CR4/5 domains of hTR. TEN, TRBD, RT, and CTE are the same as in (**B**).

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
