# Peer review of "Post-Transcriptional and Post-Translational Modifications in Telomerase Biogenesis and Recruitment to Telomeres"

_ijms, 2023, doi:10.3390/ijms24055027_

Round 1
Reviewer 1 Report
In the manuscript, Shepelev et al. summarize the recent findings on the biogenesis of telomerase components and assembly of active telomerase. The authors stress the significance of nuclear non-membrane organelles, nucleoli and Cajal bodies, in the regulation of telomerase activity and its telomere targeting. The review is an important contribution to the field. It is a comprehensive review supported by original figures and relevant references. There are some sections and paragraphs where the clarity could be improved. The comments listed below will help to clarify the meaning of the text and make it more readable for the broad scientific community.
- Lines 42-43, specify here what these two telomerase components are.
- Line 47, “modification of protein components” is unclear. If you mean TERT, it should be "protein component", if you mean TERT and telomerase accessory proteins, modify the sentence.
- Line 90, it is confusing what is “other telomerase RNP components”. In various contexts, the authors the authors use the same term “telomerase”, denoting either a two-component enzyme (TERC and TERT) or telomerase holoenzyme, with its interactors. Please, unify the definitions of telomerase and telomerase holoenzyme along the text.
- The authors review telomerase biogenesis in yeast and vertebrates as it is declared in the abstract. It would be logical to compare and discuss the similarities and differences in the crucial stages of this process in so distant species.
- It should be “Figure 3” instead of “Figure 2” in lines 257 and 295.
- The section 3.3. is confusing and hard to read. The heading and the phrases in the lines 323-326, 328-329, 333-336 should be rephrased for clarity.
- Line 318, “This motif is used…”. Do you mean that snoRNA and scaRNAs harbor this motif?
- Line 340, “Two complete complexes…” The composition of these complexes should be specified.
- Lines 336-337, the composition of “first tetramer” and “second tetramer” should be specified.
- Line 378. “H/ACA proteins”. Do you mean H/ACA RNPs or proteins that recognize H/ACA?
- Rephrase the heading of 3.4 section. PTM modifications can be applied only to proteins.
- Lines 453-457. Rephrase this paragraph for clarity.
- Lines 466, 532, 763. “nuclei” is written here instead of “nucleoli”; double-check this throughout the text.
Author Response
We are thakful to the reviewer for a very thoughtful reading of our manuscript and performed extensive revision of the manuscript in accordance to the suggestions.

Reviewer 2 Report
Overall, the review article “Post-transcriptional and post-translational modifications in telomerase biogenesis “ from Shepelev et al is a good review that covers the major discoveries and current state of understanding telomerase biogenesis in yeast and humans. I think the manuscript would be improved by some minor revisions but then should be ready for publication.
Content:
In the intro, a reference to a review article on ALT and stating that the minority of cancer cells are ALT should be added at line 34.
The sentence on line 37 starting with “Telomerase action is necessary during some steps of embryonal development” should be referenced and clarified. I don’t recall any study showing telomerase being required. Instead, my understanding is that functional telomeres are required. The telomerase KO mouse can last 5 generations for example.
Why are the potentially alternative non-telomere related activities of hTERT not discussed? For example: Sanyal S et al (full reference below) discuss sumolyation of hTERT. Seems the “noncanonical” hTERT activities warrant at least some mention in the body of the manuscript as well as in the conclusion/future directions.
Writing:
Overall, the writing is pretty good. There are some places were attention to articles “the” and other English specific editing is needed.
Couple of specific notes on the writing:.
The first sentence “Telomere length is associated with proliferative potential of the cells and telomerase elongates telomeres during whole life of organism in stem, germ cells and cells of constantly renewed tissues and it is activated in circumstances of cellular divisions during regeneration, immune response.” Is a bit of a run on sentence and should be cleaned up – perhaps by breaking into two sentences.
Line 15-16 it states “where each step should be regulated and attenuated to the cellular requirements” –attenuated should be “tuned to”
The sentence on line 430 “hTERT interacts with hTR through TRBD domain, while telomerase RNA through the CR4/5 domain and pseudoknot/template domain.” is not clear - Isn’t telomerase RNA hTR?
Relevant literature that should be considered for inclusion:
Páez-Moscoso DJ, et al. “A putative cap binding protein and the methyl phosphate capping enzyme Bin3/MePCE function in telomerase biogenesis”. Nat Commun. 2022 13(1):1067. doi: 10.1038/s41467-022-28545-9 and Porat, J., El Baidouri, M., Grigull, J. et al. The methyl phosphate capping enzyme Bmc1/Bin3 is a stable component of the fission yeast telomerase holoenzyme. Nat Commun 13, 1277 (2022). https://doi.org/10.1038/s41467-022-28985-3 - which discuss posttranscriptional aspects of telomerase RNA biogenesis in Pombe
Sanyal S et al “ SUMO E3 ligase CBX4 regulates hTERT-mediated transcription of CDH1 and promotes breast cancer cell migration and invasion.” Biochem J. 2020 ; 477(19):3803-3818. doi: 10.1042/BCJ20200359. – discusses post-translational SUMOlyation and it’s role in shifting hTERT functions
Author Response

(The authors gave the same response as above.)

Reviewer 3 Report
This review is an informative compilation and analysis of the roles of post-transcriptional and post-translational processes on each step in the biogenesis of telomerase biogenesis. As such the manuscript provides an important reference source on this topic. After the issues discussed below are addressed, the paper will be a valuable addition to the literature.
General comment:
The major criticism of this paper is that the quality of the information in the manuscript is compromised by a relatively poor use of English grammar. The sentences are too often difficult to read, with frequent run-on sentences leading to some areas of poor comprehension. Unfortunately, these errors are too extensive to address individually. A qualified individual or service expert in written English needs to help correct these problems.
Specific issues:
1. The section on telomerase delivery to the telomere omits some features of telomeric factors of shelterin as well as the roles of shelterin and ATM/ATR kinases in vertebrates. Similarly, in the description of the yeast system, the roles of Tel1/ATM, Rap1, Rif1 and Rif2 in telomerase recruitment. A brief description should be provided to show shelterin and telosome components (and their modification) fits into the scheme of PTMs fits into the scheme of other processes involved in telomerase delivery.
2. The authors often state that telomerase activity was activated or inhibited. Please state how this activity was measured (in vivo, in vitro, etc.).
3. Figure 2: Some terms are not defined or are defined inaccurately. Specifically, the use of the term shelterin in budding yeast is misleading. The structure in budding yeast (which is often termed the telosome) only shares an ortholog of Rap1. Neither Rap1 nor the Sir proteins are described in the Figure Legend or text.
4. Figure 3. Please define shelterin in the Figure legend.
Author Response

(The authors gave the same response as above.)

Round 2
Reviewer 3 Report
I complement the authors on a much-improved manuscript. The ability to read the manuscript more easily has raised several additional issues, however. These and some suggested changes in presentation are enumerated below.
Important Issues
1. In Section 2.1, you should discuss the association of the Est factors with TLC as it is for yKu. The information is shown in Figure 1, but the background needs to be discussed more fully. Also make mention of the fact that OB-domain proteins are a hallmark of an important evolutionarily conserved subclass of telomere proteins.
2. Line 94: Define Est as Every Shorter Telomere important since it was the first mutant to show telomere-related senescence.
3. Line 135: When discussion S. pombe, note that it is a fission yeast that is highly diverged from budding yeast.
4. Figure 2. The CST complex is not shown, only Stn1. Please add to figure.
5. Line 103: Define Nrd1 pathway.
6. Section 2.3
1st paragraph: define the pathway/nature of Mex67.
7. Line 227: You should mention the essential role of helicases such a Pif1 is providing accessibility to telomerase.
8. Line 229: You probably should refer to complex as Sir4-Yku80-TLC1.
9. Line 254: The Mre11 complex plays a complex regulatory role. It appears to play a structural role mediated through binding to the telomere (see Lustig 2019, Trends in Cell Biology, Towards the mechanism of yeast telomere dynamics)
10. Line 397 is unclear: In yeast, H/ACA snoRNAs are transcribed independently by RNA polymerase II and 397 are processed from a precursor, including telomerase RNA [142].
11. Line 402 is confusing, please clarify: If mature hTR is expressed 402 as an intron or RNA polymerase III is utilized only the_3_’-end of hTR carrying the H/ACA 403 motif can be detected [116].
12. Line 483 Define Cajal bodies.
13. Line 504: Rewrite the sentence: hTERT interacts with hTR through the TRBD domain, while hTR associates with TERT? the CR4/5 domain and pseudoknot/template domain.
14. Line 769 Add a brief discussion of RTEL a central component of telomere homeostasis (see Deng et al (Tzfati group) Proceedings of the National Academy of Sciences Inherited mutations in the helicase RTEL1 cause telomere dysfunction and Hoyeraal–Hreidarsson syndrom3 2013.
15. Line 843, 880 and associated Figure. You need to discuss that accessibility of the telomere is mediated through T-loop instability. This is a key step.
Parts of the Abstract and Introduction still needs some additional clarity. I have made some suggestions for simple changes below:
1. Line 14: The biogenesis of telomerase components, and their assembly and functional localization to the telomere is a complex system regulated at multiple levels.
2. Line 15 Replace “disadvantage” with “defect.”
3. Line 2021. Change to “…the development of approaches used to manipulate telomerase to influence these processes.”
4. Line 33: replace end of sentence with “and the affected cells become subject to DNA damage arrest and/or senescence.”
5. In Introduction, Line 35: place ALT pathway at end of paragraph. It will be less disruptive.
.
.
